# Agentic Data Intelligence for General Tabular Modeling

**Jun-Peng Jiang** [1] [2]  **An-Yang Ji** [1] [2]  **Jia-Yi Zhu** [1] [2]  **Han-Jia Ye** [1] [2]

## Abstract

Tabular data is one of the most common forms for organizing real-world information, supporting diverse tasks such as prediction, reasoning, querying, and generation. Despite rapid progress in tabular learning, most existing methods remain specialized for isolated problem formulations. Recent advances in large language models and agentic systems have shown increasing potential for general-purpose problem solving, offering an opportunity for unified tabular modeling. However, applying such models to tables remains challenging, as language models are often insensitive to numerical and structural information, while tabular tasks are highly heterogeneous in objectives, inputs, and evaluation protocols. In this work, we propose a general tabular agent for agentic data intelligence, TABAGENT, aiming to solve diverse table-centric tasks within a unified framework. The agent integrates rich tool-use interfaces, structured task-solving skills, and a reliable execution harness, allowing it to select appropriate actions and reasoning strategies for different tabular scenarios. Beyond standard table processing, we further endow the agent with specialized capabilities, including multimodal table understanding, table tabulation, and implicit prediction through problem decomposition. Experiments across diverse tabular tasks demonstrate that our method effectively handles heterogeneous task requirements and moves toward general tabular modeling.

## 1. introduction

Real-world applications involve diverse data-centric needs across domains such as healthcare (Lee et al., 2022; Shi et al., 2024), finance (Chen et al., 2021; Zhu et al., 2021; Chen et al., 2022b), and business (Dong et al., 2024), with tables serving as a fundamental structure for organizing heterogeneous information (Jiang et al., 2026). These needs give rise to diverse table-centric tasks, from prediction and reasoning to querying and generation, yet existing methods are typically designed as task-specific models or pipelines.

Recent advances in Large Language Models (LLMs) (Kojima et al., 2022; Yang et al., 2025; OpenAI, 2025) and agentic systems (Shi et al., 2024; Anthropic, 2025; Plaat et al., 2025) have pushed models toward more general problem-solving capabilities, including flexible reasoning, tool use, and system control (Yao et al., 2022; Qin et al., 2024; Chen et al., 2023). Although LLMs can naturally interact with tabular contexts in generation or question answering tasks, strong performance across table-centric tasks, including structured-data prediction and complex analytical reasoning, still typically requires heavy pretraining, task-specific fine-tuning, or carefully tailored adaptation (Wu et al., 2025b; Su et al., 2024; Gardner et al., 2024).

Despite their broad capabilities, applying these models and agents to tabular data still faces substantial challenges. Since LLMs are primarily trained on text, they often lack sensitivity to numerical values, exact computation, and structural relations, which are crucial for tabular reasoning (Jiang et al., 2025; Sun et al., 2025). Meanwhile, the heterogeneity of tables and the specificity of table-centric tasks make it difficult for task-specific training or adaptation to generalize across different problem formulations (Gardner et al., 2024; Wen et al., 2024; Ji et al., 2026). Moreover, real-world tables often appear not only as clean digital records, but also in documents, screenshots, and other multimodal contexts, requiring models to jointly handle visual layouts and structured semantics (Jiang et al.; Jiang, 2026).

To address these challenges, we propose a general tabular agent TABAGENT for agentic data intelligence, aiming to solve diverse table-centric tasks within a unified framework. To mitigate the limited numerical and structural sensitivity of language models, we equip the agent with rich tool-use interfaces, including data analysis, predictive modeling, SQL, and visualization tools. These tools enable the agent to inspect, manipulate, and reason over entire tables through executable operations, rather than relying only on textual input. To further handle heterogeneous task requirements, we design structured skills for preprocessing, analysis, pre-

[1]School of Artificial Intelligence, Nanjing University [2]National Key Laboratory for Novel Software Technology, Nanjing University. Correspondence to: Han-Jia Ye <yehj@lamda.nju.edu.cn>.

*Proceedings of the $2^{nd}$ ICML Workshop on Foundation Models for Structured Data*, Seoul, South Korea. Copyright 2026 by the author(s).

diction, visualization, and question answering. A reliable execution harness then coordinates task interpretation, tool invocation, intermediate verification, and final response generation, allowing the agent to select appropriate actions and reasoning strategies across different tabular scenarios.

Furthermore, to make TABAGENT applicable to broader and more realistic tabular scenarios, we extend it beyond standard table processing with several specialized capabilities. We first equip it with multimodal table understanding, enabling it to handle tables embedded in documents, screenshots, and other visual contexts. We then introduce table tabulation, which transforms unstructured or semi-structured information into table-like representations for prediction. Finally, we enable implicit prediction through problem decomposition, allowing the agent to infer hidden predictive objectives from natural language requests and construct suitable modeling workflows. Extensive experiments across diverse tabular tasks demonstrate the comprehensiveness and generality of TABAGENT. Our main contributions are summarized as follows:

- We identify the diversity and heterogeneity of table-centric tasks, and motivate the need for general tabular modeling as a practical direction beyond isolated task-specific methods.
- We introduce TABAGENT, a general tabular agent that combines tool use, structured skills, and execution harnesses to address diverse tabular tasks, further enhanced with multimodal table understanding, table tabulation, and implicit prediction.
- We evaluate TABAGENT on a broad range of tabular tasks, showing that it can effectively handle heterogeneous task requirements and move toward general agentic data intelligence for tables.

## 2. Related Work

### 2.1. Tabular-Centric Tasks

A growing body of benchmarks has evaluated table-centric reasoning from different perspectives. Spider (Yu et al., 2018; Lei et al., 2024) extends NL2SQL evaluation to realistic enterprise database querying. TableBench (Wu et al., 2025c) evaluates comprehensive TableQA abilities under multiple reasoning paradigms, such as table chain-of-thought, standard chain-of-thought, and program-of-thought. TableEval (Zhu et al., 2025) considers question answering over multi-structured tables, requiring models to handle heterogeneous table organizations. LongTableBench (Li et al., 2025) focuses on long-context table reasoning across domains. RealHiTBench (Wu et al., 2025a) targets hierarchical tables with complex multi-level structures rather than flat tables. TopBench (Ji et al., 2026) introduces implicit predictive TableQA, requiring models to identify latent pre-

diction goals and conduct prediction-oriented reasoning. These benchmarks collectively reveal the diversity of table-centric tasks and highlight the limitations of developing separate models or pipelines for each setting. Moreover, MMTU (Jiang et al., 2025) introduces a multimodal benchmark for image table understanding from different aspects.

### 2.2. LLMs and Agents

Large language models have evolved into more agentic systems that can reason, act, and interact with external environments. Prior work improves LLM reasoning through step-by-step prompting, self-consistency, program-aided reasoning, and tree-structured search (Wei et al., 2022; Kojima et al., 2022; Wang et al., 2023; Chen et al., 2022a; Gao et al., 2023; Yao et al., 2023), while self-reflection and iterative refinement enable models to evaluate and revise their outputs (Madaan et al., 2023; Shinn et al., 2024; Zhou et al., 2024; Yang et al., 2024). Beyond reasoning, tool-augmented agents connect LLMs with external APIs, calculators, code interpreters, search engines and databases, grounding model decisions in executable computation and retrieved information (Lewis et al., 2020; Yao et al., 2022; Schick et al., 2023; Qin et al., 2024; Patil et al., 2023; Zhuang et al., 2023). Recent studies further explore multi-agent frameworks, role-based collaboration, and agent communication for solving complex tasks (Wu et al., 2023; Li et al., 2023; Chen et al., 2023; Chan et al., 2023; Gao et al., 2024). These advances provide a general foundation for agentic problem solving, but most existing agents target general language, web, or tool-use scenarios rather than heterogeneous table-centric tasks that require numerical sensitivity, structural understanding, and reliable tabular operations.

## 3. Method

### 3.1. Preliminary

Figure 1 illustrates the overall architecture of TABAGENT. Given a user query and session-level table context, the system first routes the request to suitable table-centric operations, then invokes executable tools through an execution harness, and finally verifies and synthesizes the tool outputs into the final response. Given a session $\mathcal{S} = (\mathcal{T}, \mathcal{H})$ of uploaded tables or table images and dialogue history, TABAGENT maps a user query $q$ to an executable tabular workflow. The key difficulty is that table-centric requests are heterogeneous: some require exact computation over observed rows, whereas others require data cleaning, visualisation, multi-table transformation, or prediction over unseen outcomes. TABAGENT therefore treats table intelligence as a routing-and-execution problem rather than a pure text-generation problem.

Each turn produces a structured tool output $r = (y, m, p, a)$,

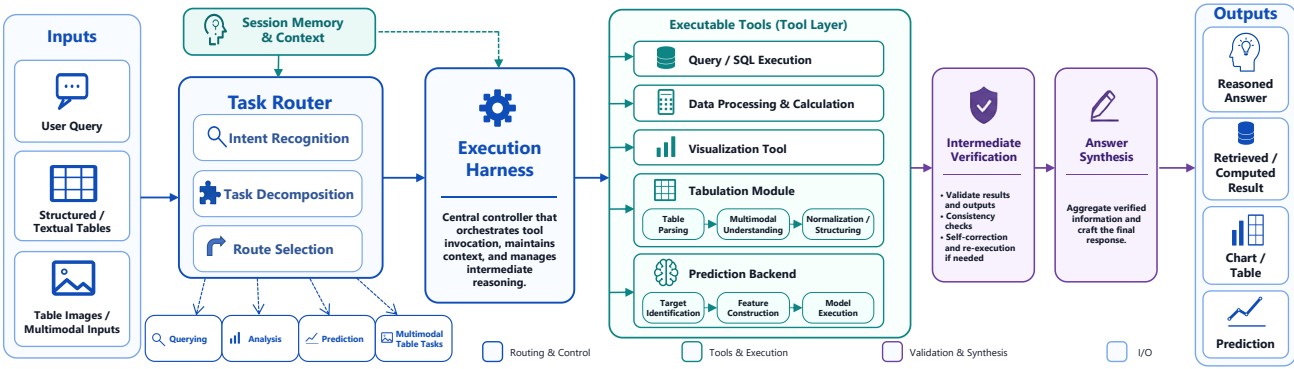

*Figure 1.* Architecture of TABAGENT for general tabular modeling. The framework routes table-centric requests to executable tools and specialized modules for tabulation, multimodal understanding, and implicit prediction, then synthesizes verified outputs into final answers. This design turns heterogeneous table reasoning into a unified routing-and-execution process.

where $y$ is the textual answer, $m$ is metadata, $p$ is a row-level preview, and $a$ denotes generated artefacts. This representation lets the model reason over executable results instead of raw table text, directly addressing the numerical and structural weakness of language-only models.

### 3.2. TABAGENT

TABAGENT contains three components. A *session memory* stores uploaded assets, the active table, and lightweight metadata such as schema, missingness, and previews, enabling iterative workflows and multi-table context across turns. A *task router* maps $q$ to one primary intent among profiling, querying, preprocessing, visualisation, prediction, and tabulation, together with secondary intents when needed. The router uses task-specific lexical cues and explicit boosts for latent prediction and prediction-oriented table construction. A *tool layer* then executes the selected operation through local tabular functions.

The tool layer is designed to cover the main failure modes of LLMs on tables. For structural understanding, the profiling tool summarises table purpose, field semantics, numeric statistics, and quality issues. For exact reasoning, the query tool executes SQL-style analysis. For data preparation, the preprocessing tool diagnoses missingness, duplicates, and IQR-based outliers, and applies adaptive imputation rules. For visual analysis, the visualisation tool generates executable plots instead of verbal summaries. Together, these tools ground the agent in explicit computation.

To handle hidden predictive objectives, TABAGENT introduces an implicit-prediction module. When a request asks for an unseen value, a likely outcome, or the best decision, the system recasts it as prediction rather than retrieval. It infers the target column and feature set from the query and schema, extracts an inference row from natural language when necessary, and selects a baseline model according to table scale and feature type: TabPFN for small tables,

LightGBM or XGBoost for larger tables, and CatBoost for categorical-heavy data, with tree-based fallbacks otherwise. This decomposition converts ambiguous TableQA into explicit tabular modelling.

To support realistic relational data, TABAGENT further includes a tabulation module. Given multiple source tables, it infers an entity key, resolves bridge keys when the entity key is missing, aggregates event records into entity-level features, and materialises a prediction-ready wide table. Numeric columns contribute summary statistics and latest values. Categorical columns contribute the latest values, modes, and distinct-count features. The generated table is registered back into the session for downstream prediction. Finally, table images are also supported conservatively through direct QA. This extends coverage to multimodal inputs without overstating structured extraction ability.

## 4. Experiment

### 4.1. Experimental Setup

We evaluate TABAGENT on seven benchmarks covering executable NL2SQL, general TableQA, long-table reasoning, hierarchical analysis, and implicit predictive TableQA: Spider 2.0-Snow (Lei et al., 2024), TableBench (Wu et al., 2025c), TableEval (Zhu et al., 2025), LongTableBench (Li et al., 2025), RealHiTBench (Wu et al., 2025a), TOP-BENCH (Ji et al., 2026), and MMTU (Jiang et al., 2025). We use each benchmark's official setting and metric, and compare against reported strong foundation-model and agent baselines. This suite matches the three challenges: numerical and structural reasoning, task heterogeneity, and broader multimodal or prediction-oriented table use.

### 4.2. Main Results

Table 1 shows that TABAGENT performs strongly across highly different tasks. On Spider 2.0-Snow, it reaches 48.9

*Table 1.* Model performance comparison across textual table-centric benchmarks. Compared with representative foundation-model and agent baselines, TABAGENT achieves the best or highly competitive performance across heterogeneous evaluation settings. The results highlight the value of executable tool use and structured task routing for improving numerical and structural reasoning across diverse table tasks. (– indicates not directly reported.)

| Model | Spider | TBench | TEval | LTBench | | | RealHiT | | | | | TOPBENCH | | | | |
|---|---|---|---|---|---|---|---|---|---|---|---|---|---|---|---|---|
| | *Succ.* | *Avg* | *Avg* | *Tsk* | *Fmt* | *128K* | *FC* | *NR* | *SC* | *DA* | *CG* | *SP* | *DM* | *TE* | *RC* | *RN↓* |
| DeepSeek-R1 | – | – | 82.46 | 62.82 | 65.55 | 49.45 | 79.45 | 72.54 | 84.62 | **79.55** | 7.14 | – | – | – | – | – |
| DeepSeek-V3.2-T | – | – | – | – | – | – | – | – | – | – | – | 0.61 | 0.58 | **0.65** | 0.38 | 0.35 |
| Qwen2.5-7B-Instruct | – | 22.14 | 59.60 | 32.05 | 33.65 | 25.95 | 38.39 | 19.75 | 44.81 | 40.17 | 15.58 | – | – | – | – | – |
| Qwen2.5-72B-Instruct | – | 48.79 | 74.23 | 49.94 | 51.42 | 39.36 | 62.15 | 39.23 | 68.34 | 68.45 | 14.29 | – | – | – | – | – |
| Qwen3-Thinking | – | 52.45 | – | 34.94 | 36.27 | 26.57 | – | – | – | – | – | 0.57 | 0.57 | 0.53 | 0.42 | 0.46 |
| GPT4o | 10.1 | 45.18 | 78.79 | 65.60 | 64.16 | **54.92** | 68.97 | 50.12 | 71.14 | 73.37 | 20.13 | – | – | – | – | – |
| GPT5.2 | – | – | – | – | – | – | – | – | – | – | – | 0.60 | 0.55 | **0.65** | 0.38 | 0.41 |
| Claude 3.5 Sonnet | – | – | 83.32 | 60.63 | 59.60 | 47.84 | – | – | – | – | – | – | – | – | – | – |
| Claude Sonnet 4.5 | – | – | – | – | – | – | – | – | – | – | – | 0.64 | 0.56 | 0.63 | 0.55 | 0.31 |
| Gemini 1.5 Pro | – | – | – | – | – | – | 66.14 | 43.74 | 69.71 | 70.72 | 9.74 | – | – | – | – | – |
| Gemini 3 Flash | – | – | – | – | – | – | – | – | – | – | – | 0.66 | **0.65** | **0.65** | 0.58 | 0.30 |
| TABAGENT | **48.9** | **53.41** | **86.93** | **74.56** | **75.10** | 52.98 | **89.81** | **90.12** | **88.42** | 77.03 | **90.13** | **0.71** | 0.54 | 0.59 | **0.70** | **0.22** |

*Table 2.* Performance comparison on the MMTU multimodal benchmark. TABAGENT achieves strong performance on multimodal table tasks, showing that the proposed framework can generalise beyond textual inputs.

| Model | MMTU | | | |
|---|---|---|---|---|
| | *IE* | *RC* | *CC* | *CR* |
| DeepSeek-VL2 | 0.90 | 0.55 | 0.33 | 0.24 |
| Qwen2-VL | 0.93 | 0.71 | 0.38 | 0.38 |
| TABAGENT | **0.95** | **0.84** | **0.49** | **0.76** |

success rate, substantially above the listed general-model baselines. On TableEval, it achieves the best average score of 86.93. On LongTableBench, it obtains the best TaskAvg and FmtAvg, indicating that explicit routing and tool execution improve robustness on long and heterogeneous tables. On RealHiTBench, it improves all listed text-input metrics, suggesting that structured intermediate analysis is effective for complex table organisation.

The advantage is especially clear on TOPBENCH. TABAGENT achieves the best single-point prediction score and the strongest ranking results, including the lowest ranking-regression error. This result supports the central design choice of the method: implicit predictive queries should be decomposed into target identification, feature construction, and model execution, rather than handled as ordinary retrieval-style TableQA.

### 4.3. Discussion

**Performance on Textual QA Settings.** The overall pattern is consistent with the method design. Gains on Spider 2.0-Snow, TableEval, and RealHiTBench reflect the value of replacing free-form reasoning with executable operations. Gains on TOPBENCH reflect the value of explicit latent-prediction routing. The weaker result on the

program-of-thought metric in TableBench suggests a trade-off: TABAGENT prioritises reliable external execution over long internal derivation. More generally, the results indicate that a unified agentic framework can remain competitive across diverse table tasks without task-specific retraining, while still leaving room for stronger multimodal parsing and more advanced predictive backends.

**Performance on Multimodal Settings.** The multimodal results show a similar pattern. The most pronounced gains are observed on CR, with notable improvements on RC and CC, suggesting that TABAGENT is particularly effective at handling complex reasoning over multimodal inputs through execution-based workflows. In contrast, the marginal gains on IE indicate that basic perception is not the primary bottleneck. These results reflect the value of structured execution and intermediate verification in supporting complex reasoning. Overall, TABAGENT demonstrates robust performance in multimodal settings, while further improvements remain possible in cross-modal reasoning and alignment.

## 5. Conclusion

We presented TABAGENT, a general tabular agent for heterogeneous table-centric tasks. The method addresses three core difficulties in data intelligence: weak numerical and structural grounding in language models, large variation across task types, and the need to handle broader settings such as implicit prediction and multi-table tabulation. To this end, TABAGENT combines task routing, executable tabular tools, implicit-prediction decomposition, and entity-level tabulation in a unified framework. Experiments across seven benchmarks show that this design is effective over substantially different table tasks, with especially strong gains on benchmarks that require exact execution or latent predictive reasoning. These results suggest that general tabular modelling is a promising setting for agentic methods.

## Acknowledgements

This work is partially supported by National Key R&D Program of China (2024YFE0202800), Basic Research Program of Jiangsu under Grants (BK20253021), NSFC (62522605, 62376118, 62522216, 62402408), Hong Kong SAR Research Grants Council (RGC) Early Career Scheme (26208924), Hong Kong SAR Research Grants Council (RGC) General Research Fund (16219025), the Fundamental and Interdisciplinary Disciplines Breakthrough Plan of the Ministry of Education of China (No. JYB2025XDXM118), the "111 Center" (No. B26023), the Collaborative Innovation Center of Novel Software Technology and Industrialization.

## Impact Statement

This work aims to advance general-purpose machine learning systems for tabular data. By combining language-model agents with executable tools, task routing, multimodal table understanding, and implicit prediction workflows, TABAGENT may reduce the need for task-specific tabular pipelines and help users perform table querying, analysis, and prediction in a unified interface. However, tabular data often contains sensitive or domain-specific information, and outputs from such systems should not be used in high-stakes settings without proper validation, privacy protection, and human oversight.

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

# A. Benchmark Details

We summarize the benchmarks used in our evaluation in Table 3. These benchmarks cover executable querying, multi-structured TableQA, long-context reasoning, hierarchical table analysis, and implicit predictive TableQA. For each benchmark, we keep the official setting and evaluation metrics used in the corresponding source.

*Table 3.* Metadata of the benchmarks used in our evaluation. The table reports the task type, dataset scale, official setting, and metrics retained for comparison.

| Benchmark | Type | Size | Official setting retained | Metrics (Side-by-side in matrices) |
|---|---|---|---|---|
| Spider 2.0-Snow | Enterprise NL2SQL | 547 examples | Official website baseline | Success rate (%) |
| TableBench | Comprehensive TableQA | 886 samples, 18 fields | Main results table | TCoT, SCoT, PoT |
| TableEval | Multi-structured TableQA | 5,422 QA; 617 tables | Main results table | Avg SEAT |
| LongTableBench | Long-context table reasoning | 5,950 QA; 18 domains | Representative results table | TaskAvg, FmtAvg, 128K |
| RealHiTBench | Hierarchical table analysis | 708 tables; 3,752 QA | Main table, Text input | FC-F1, NR-F1, SC-F1, DA-GEval, CG-P@1 |
| TOPBENCH | Implicit predictive TableQA | 779 samples; 35 tables | Main results table | SP, DM, TE, RankCls, RankRegNMAE↓ |

# B. Additional Implementation Details

This section describes the implementation settings used in our experiments.

**Runtime.** TABAGENT is implemented in Python. When the corresponding Claude environment is available, the system uses `claude-agent-sdk` as the primary orchestration runtime. Table-centric capabilities are implemented as in-process executable tools and registered to the runtime through an MCP-compatible interface. The agent layer handles planning and response synthesis, while profiling, querying, preprocessing, visualisation, prediction, and tabulation are delegated to local tool procedures.

**Model configuration.** The backbone model is specified by runtime configuration. In our implementation, the model name is read from `TABAGENT_MODEL`, with `claude-sonnet-4-5` as the default value. At inference time, the runtime combines the system prompt, session context, routing signals, and skill-specific guidance before exposing the tabular tools to the agent through a constrained SDK tool interface. For robustness, the system also includes a local execution mode that preserves session state and can invoke the tool layer directly when full agent orchestration is unavailable. This mode is used as a fallback path rather than as an equivalent replacement for the full agent runtime.

**Session state.** TABAGENT supports both table-free dialogue and table-grounded multi-turn interaction. For each session, the storage layer maintains uploaded assets, the currently active asset, lightweight metadata, recent previews, and generated artefacts. For structured tables, the persisted metadata include row and column counts, column names, data types, missing-value ratios, unique-value counts, a caption-style summary inferred from the file name, and a short preview of table rows. These summaries provide compact grounding context for routing and tool selection without serialising the full table into the prompt.

**Data and artifact handling.** A single session may contain multiple uploaded assets. The most recently uploaded asset is used as the default active item, while previously uploaded assets remain available for later operations. The supported structured input formats are `csv`, `tsv`, `txt`, `xlsx`, `xls`, `json`, and `jsonl`. For image-based table inputs, the runtime keeps the uploaded image as an asset and passes visual evidence directly to the multimodal model, rather than first converting it into a structured DataFrame.

Runtime files are stored under a configurable working directory, with `.tabagent_runtime` as the default. Session-local metadata are stored under `sessions/`, while generated plots and exported tables are stored under `artifacts/`.

**Tool outputs.** Each executable tool returns a structured `ToolResult`. It contains a textual summary, metadata, an optional tabular preview, optional artefact paths, generated code when applicable, and an error status. Querying is implemented with DuckDB over in-memory DataFrame objects, while visualisation saves Matplotlib figures as PNG artefacts. Prediction and

tabulation tools are invoked when the router identifies prediction-oriented or multi-table requests. Generated artefacts and transformed tables are written back to the session state, so later turns can refer to previous intermediate results.

## C. Task Routing and Skill Use

We provide further details on the task router and the skill prompts used by TABAGENT. The router converts a user request into an executable table-centric workflow before the final answer is generated.

**Route decision.** Given a user request and the current session state, the router produces a route decision that contains the primary intent, optional secondary intents, recommended tools, and matched table columns. The current router is heuristic rather than learned. It scores lexical cues against intent-specific vocabularies and also checks whether the user utterance explicitly mentions column names from the active table. In this way, the route decision depends on both the natural-language request and the available table context.

**Routing policy.** The router covers profiling, querying, preprocessing, visualisation, prediction, tabulation, and causal boundary cases. When a query asks for aggregation, filtering, comparison, or direct inspection, the system routes it to the query or profiling pathway. When the query asks for cleaning, type conversion, duplicate removal, or missing-value handling, it is routed to preprocessing. Visual requests are routed to the visualisation tool. If no strong cue is detected, the system defaults conservatively to the query pathway.

**Prediction and tabulation cues.** Two score adjustments are used for cases that are central to TABAGENT. Requests referring to unseen values, future outcomes, likely outcomes, or decision recommendations are biased toward the prediction intent, which supports implicit prediction rather than direct lookup. Requests that mention constructing a wide table, feature table, or prediction-ready table are biased toward tabulation. These rules allow the system to handle requests that require target inference or entity-level table construction before answering.

*Table 4.* Intent categories used by the heuristic router. Each intent corresponds to a major executable workflow supported by our TABAGENT implementation.

| Intent | Typical request | Main operation |
|---|---|---|
| Profiling | Describe the table, inspect fields, or summarize missing values | Compute schema summaries, column profiles, quality flags, and previews |
| Querying | Count, filter, aggregate, or compare table records | Execute explicit SQL or answer lightweight metadata-oriented QA |
| Preprocessing | Clean missing values, remove duplicates, detect outliers, or convert types | Apply table transformations and update the session state |
| Visualisation | Draw trends, distributions, or group comparisons | Generate local chart artefacts from selected columns |
| Prediction | Predict an unseen value, estimate an outcome, or make a decision | Infer the supervised task and invoke a predictive model |
| Tabulation | Construct a table from multiple tables before analysis or prediction | Materialize an entity-level wide table for downstream use |

Prediction-oriented requests are therefore handled differently from direct table lookup questions. Similarly, relational or multi-table requests are routed to tabulation when an entity-level representation must be constructed before downstream analysis or prediction. Causal requests are detected as boundary cases and handled conservatively, since causal analysis is outside the scope of the evaluated backend.

**Skill prompts.** Skill prompts are loaded according to the routed intent. They provide task-specific guidance before tool calls are made, for example by encouraging schema inspection, executable aggregation or transformation, and decomposition of latent predictive requests into target identification, feature construction, model execution, and response synthesis.

## D. Metric and Acronym Glossary

For ease of reference, we summarize the benchmark acronyms and metric abbreviations used in the main result tables.

*Table 5.* Glossary of benchmark and metric abbreviations used in the result tables.

| Abbreviation | Meaning | Reference |
|---|---|---|
| Spider 2.0-Snow / Spider | Spider 2.0-Snow benchmark for enterprise-style NL2SQL evaluation | Spider 2.0-Snow (Lei et al., 2024) |
| TableBench / TBench | TableBench, a comprehensive and complex benchmark for TableQA | TableBench (Wu et al., 2025c) |
| TableEval / TEval | TableEval, a real-world benchmark for complex, multilingual, and multi-structured TableQA | TableEval (Zhu et al., 2025) |
| LongTableBench / LTBench | LongTableBench, a benchmark for long-context table reasoning | LongTableBench (Li et al., 2025) |
| RealHiT / RealHiTBench | RealHiTBench, the Realistic Hierarchical Table Benchmark | RealHiTBench (Wu et al., 2025a) |
| TOPBENCH | TopBench, a benchmark for implicit prediction and reasoning in TableQA | TOPBENCH (Ji et al., 2026) |
| MMTU | Multimodal Table Understanding benchmark used for image-based table evaluation | MMTU (Jiang et al., 2025) |
| Succ. | Success rate | Spider 2.0-Snow official metric |
| TCoT | Table Chain-of-Thought | TableBench (Wu et al., 2025c) |
| SCoT | Standard Chain-of-Thought | TableBench (Wu et al., 2025c) |
| PoT | Program-of-Thought | TableBench (Wu et al., 2025c) |
| SEAT | The sub-question-level semantic alignment metric proposed by TableEval | TableEval (Zhu et al., 2025) |
| TaskAvg / Tsk | Task average | LongTableBench official reporting |
| FmtAvg / Fmt | Format average | LongTableBench official reporting |
| 128K | Score under the 128K-context evaluation setting | LongTableBench official reporting |
| FC-F1 / FC | Fact Checking F1 | RealHiTBench (Wu et al., 2025a) |
| NR-F1 / NR | Numerical Reasoning F1 | RealHiTBench (Wu et al., 2025a) |
| SC-F1 / SC | Structure Comprehending F1 | RealHiTBench (Wu et al., 2025a) |
| DA-GEval / DA | Data Analysis scored with G-Eval | RealHiTBench (Wu et al., 2025a) |
| CG-P@1 / CG | Chart Generation scored with Pass@1 | RealHiTBench (Wu et al., 2025a) |
| SP | Single-Point Prediction | TOPBENCH (Ji et al., 2026) |
| DM | Decision Making | TOPBENCH (Ji et al., 2026) |
| TE | Treatment Effect Analysis | TOPBENCH (Ji et al., 2026) |
| RankCls / RC | Ranking-classification score in the ranking/filtering sub-evaluation | TOPBENCH (Ji et al., 2026) |
| RankRegNMAE / RN | Ranking-regression normalized mean absolute error | TOPBENCH (Ji et al., 2026) |
| MMTU-IE | Understanding individual elements | MMTU (Jiang et al., 2025) |
| MMTU-RC | Interpreting rows or columns | MMTU (Jiang et al., 2025) |
| MMTU-CC | Comprehending compositional conditions | MMTU (Jiang et al., 2025) |
| MMTU-CR | Performing basic calculations or reasoning over table values | MMTU (Jiang et al., 2025) |

