# OpenReview forum: "Agentic Data Intelligence for General Tabular Modeling"
_ICML.cc/2026/Workshop/FMSD — FMSD @ ICML 2026 Poster_

### Official Review · Reviewer_SpDQ · 2026-05-20
**Agentic Tabular Modelling, Very Practical**

**Rating:** 6
**Confidence:** 3

**Review:**

Summary of Contributions
TabAgent is an agentic framework that routes diverse table-centric tasks (QA, prediction, querying, visualization, tabulation) to executable tools (SQL, TabPFN, LightGBM, etc.) via an LLM with structured task routing. Prediction-related tasks eventually have a model trained (TabPFN, XGBoost, or CatBoost). Evaluated on seven benchmarks, it achieves strong performance across heterogeneous settings.

Strengths
- Very practical; unifies tabular task pipelines under one agent.
- Broad evaluation across seven diverse benchmarks with strong results.
- Empirically convincing that the framework handles heterogeneous tasks.

Weaknesses
- Some table columns use unexpanded acronyms, hurting readability.
- TabAgent orchestrates and trained actual tabular models while baselines are pure LLMs. An ablation of some kind would be beneficial, or else, this feels like a bit of an unfair comparison.

Suggestions
- Expand acronyms and improve table labeling, and share more details of the TabAgent setup (the code, or what LLM was used)

---

### Official Review · Reviewer_oifL · 2026-05-21
**An Agentic Architectural Design for Series of Tabular Data Tasks**

**Rating:** 7
**Confidence:** 5

**Review:**

### Summary

This paper proposes a unified, general-purpose agentic framework tailored for diverse tabular tasks. The system provides agents with table-native skills and a dedicated tool layer, utilizing an architecture that integrates session memory, a task router, and execution tools.

### Strengths

- Unified & Intuitive Architecture: The framework provides an approach to various tabular tasks, spanning preprocessing, modeling, and verification. Furthermore, the strategy of routing tasks to different models based on scale is very intuitive. Therefore, the envisioned system is likely to inspire discussion within the community.
- Good Evaluation Coverage: The experimental design covers both analytical and predictive tabular tasks.

### Areas for Improvement

- Clarity in Experimental Baselines: The preliminary results table currently compares the proposed agentic system directly against foundation models, which may not be entirely fair. It would be more valuable to compare the system against existing agentic frameworks, such as a ReAct-based agent. Although the caption indicates that the comparison involves both agents and foundation models but in the table, there is only FMs. If we look at the Spider2.0-snow leaderboard online, there are results using various agents. Elaborating on the exact experimental setup would improve clarity and readability.
- Scope of Data Scenarios: The evaluation would be strengthened by expanding the scope to include relational data scenarios.
- Architectural Details: The design of the task router is a critical component of the overall performance. The paper would benefit more from an in-depth discussion of the design details.

---

### Official Review · Reviewer_Pmrc · 2026-05-21

**Rating:** 5
**Confidence:** 3

**Review:**

**Summary**

The paper introduces an LLM-agent tailored for a wide-ranging set of tabular learning tasks. The agent is compared against prior work on a wide range of benchmarks, demonstrating state-of-the-art performance

**Strength**

The work on combining many tabular tasks and solving them with a single scaffold and an LLM is novel and timely. This work is relevant for the workshop.

**Areas for improvement**

Currently there are important details missing regarding the empirical results. The underlying LLM is not specified, is it the case that the scaffold helps, or is it the case that its just a better model equiped with some "code" tool? Follow-up: there is no baseline of same LLM, simpler tools -- that would demonstrate the need for a special scaffold.

**Justification of Score**

Overall, I think that the paper is good and well executed and nicely fits the workshop, modulo the important details regarding the setup and an important missing ablation. I'm okay with the paper being accepted, but am cautious because of the lacking emperical evidence in the current submission -- hence a weak reject (but not a strong opinion: even if the result is just from using the SoTA LLM, it is still interesting, given the degree of performance improvements over prior work and its uniformity)